# Saliency, Scale and Information:
# Towards a Unifying Theory

**Shafin Rahman**
Department of Computer Science
University of Manitoba
shafin109@gmail.com

**Neil D.B. Bruce**
Department of Computer Science
University of Manitoba
bruce@cs.umanitoba.ca

## Abstract

In this paper we present a definition for visual saliency grounded in information theory. This proposal is shown to relate to a variety of classic research contributions in scale-space theory, interest point detection, bilateral filtering, and to existing models of visual saliency. Based on the proposed definition of visual saliency, we demonstrate results competitive with the state-of-the art for both prediction of human fixations, and segmentation of salient objects. We also characterize different properties of this model including robustness to image transformations, and extension to a wide range of other data types with 3D mesh models serving as an example. Finally, we relate this proposal more generally to the role of saliency computation in visual information processing and draw connections to putative mechanisms for saliency computation in human vision.

## 1   Introduction

Many models of visual saliency have been proposed in the last decade with differences in defining principles and also divergent objectives. The motivation for these models is divided among several distinct but related problems including human fixation prediction, salient object segmentation, and more general measures of *objectness*. Models also vary in intent and range from hypotheses for saliency computation in human visual cortex to those motivated exclusively by applications in computer vision. At a high level the notion of saliency seems relatively straightforward and characterized by patterns that stand out from their context according to unique colors, striking patterns, discontinuities in structure, or more generally, figure against ground. While this is a seemingly simplistic concept, the relative importance of defining principles of a model, and fine grained implementation details in determining output remains obscured. Given similarities in the motivation for different models, there is also value in considering how different definitions of saliency relate to each other while also giving careful consideration to parallels to related concepts in biological and computer vision.

The characterization sought by models of visual saliency is reminiscent ideas expressed throughout seminal work in computer vision. For example, early work in scale-space theory includes emphasis on the importance of extrema in structure expressed across scale-space as an indicator of potentially important image content [1, 2]. Related efforts grounded in information theory that venture closer to *modern* notions of saliency include Kadir and Brady [3] and Jagersand's [4] analysis of interaction between scale and local entropy in defining relevant image content. These concepts have played a significant role in techniques for affine invariant keypoint matching [5], but have received less attention in the direct prediction of saliency. Information theoretic models are found in the literature directly addressing saliency prediction for determining gaze points or proto-objects. A prominent example of this is the AIM model wherein saliency is based directly on measuring the self-information of image patterns [6]. Alternative information theoretic definitions have been pro-

posed [7, 8] including numerous models based on measures of redundancy or compressibility that are strongly related to information theoretic concepts given common roots in communication theory.

In this paper, we present a relatively simple information theoretic definition of saliency that is shown to have strong ties to a number of classic concepts in the computer vision and visual saliency literature. Beyond a specific model, this also serves to establish formalism for characterizing relationships between scale, information and saliency. This analysis also hints at the relative importance of fine grained implementation details in differentiating performance across models that employ disparate, but strongly related definitions of visual salience. The balance of the paper is structured as follows: In section 2 we outline the principle for visual saliency computation proposed in this paper defined by maxima in information scale-space (MISS). In section 3 we demonstrate different characteristics of the proposed metric, and performance on standard benchmarks. Finally, section 4 summarizes main points of this paper, and includes discussion of broader implications.

## 2    Maxima in Information Scale-Space (MISS)

In the following, we present a general definition of saliency that is strongly related to prior work discussed in section 1. In short, according to our proposal, saliency corresponds to maxima in information-scale space (MISS). The description of MISS follows, and is accompanied by more specific discussion of related concepts in computer vision and visual saliency research.

Let us first assume that the saliency of statistics that define a local region of an image are a function of the rarity (likelihood) of such statistics. We'll further assume without loss of generality that these local statistics correspond to pixel intensities.

The likelihood of observing a pixel at position $p$ with intensity $I_p$ in an image based on the global statistics, is given by the frequency of intensity $I_p$ relative to the total number of pixels (i.e. a normalized histogram lookup). This may be expressed as follows: $H(I_p) = \sum_{q \in S} \delta(I_q - I_p)/|S|$ with $\delta$ the Dirac delta function. One may generalize this expression to a non-parametric (kernel) density estimate: $H(I_p) = \sum_{q \in S} G_{\sigma_i}(I_p - I_q)$ where $G_{\sigma_i}$ corresponds to a kernel function (assumed to be Gaussian in this case). This may be viewed as either smoothing the intensity histogram, or applying a density estimate that is more robust to low sample density[1]. In practice, the proximity of pixels to one another is also relevant. Filtering operations applied to images are typically local in their extent, and the correlation among pixel values inversely proportional to the spatial distance between them. Adding a local spatial weighting to the likelihood estimate such that nearby pixels have a stronger influence, the expression is as follows:

$$H(I_p) = \sum_{q \in S} G_{\sigma_b}(||p - q||) G_{\sigma_i}(||I_p - I_q||) \tag{1}$$

This constitutes a locally weighted likelihood estimate of intensity values based on pixels in the surround.

Having established the expression in equation 1, we shift to discussion of scale-space theory. In traditional scale-space theory the scale-space representation $L(x, y; t)$ is defined by convolution of an image $f(x, y)$ with a Gaussian kernel $g(x, y)$ such that $L(x, y; t) = g(., ., t) * f(., .)$ with $t$ the variance of a Gaussian filter. Scale-space features are often derived from the family of Gaussian derivatives defined by $L_{x^m y^n}(., .; t) = \delta_{x^m y^n} g(., ., t) * f(., .)$ with differential invariants produced by combining Gaussian derivatives of different orders in a weighted combination. An important concept in scale-space theory is the notion that scale selection or the size and position of relevant structure in the data, is related to the scale at which features (e.g. normalized derivatives) assume a maximum value. This consideration forms the basis for early definitions of saliency which derive a measure of saliency corresponding to the scale at which local entropy is maximal. This point is revisited later in this section.

The scale-space representation may also be defined as the solution to the heat equation: $\frac{\delta I}{\delta t} = \Delta I = I_{xx} + I_{yy}$ which may be rewritten as $G[I]_p - I \approx \Delta I$ where $G[I]_p = \int_S G_{\sigma_s} I_q dq$ and $S$ the local

spatial support. This expression is the solution to the heat equation when $\sigma_s = \sqrt{2t}$. This corresponds to a diffusion process that is isotropic. There are also a variety of operations in image analysis and filtering that correspond to a more general process of *anisotropic diffusion*. One prominent example is the that proposed by Perona and Malik [9] that implements edge preserving smoothing. A similar process is captured by the Yaroslavsky filter: $Y[I]_p = \frac{1}{C(p)} \int_{B_{\sigma_S}} G_{\sigma_r}(||I_p - I_q||) I_q dq$ [10] with $B_{\sigma_S}$ reflecting the spatial range of the filter. The difference between these techniques and an isotropic diffusion process is that relative intensity values among local pixels determine the degree of diffusion (or weighted local sampling).

The Yaroslavsky filter may be shown to be a special case of the more general bilateral filter corresponding to a step-function for the spatial weight factor [11]: $B[I]_p = \frac{1}{W_p} \sum_{q \in S} G_{\sigma_b}(||p - q||) G_{\sigma_i}(||I_p - I_q||) I_q$ with $W_p = \sum_{q \in S} G_{\sigma_b}(||p - q||) G_{\sigma_i}(||I_p - I_q||)$.

In the same manner that selection of scale-space extrema defined by an isotropic diffusion process carries value in characterizing relevant image content and scale, we propose to consider scale-space extrema that carry a relationship to an anisotropic diffusion process.

Note that the normalization term $W_p$ appearing in the equation for the bilateral filter is equivalent to the expression appearing in equation 1. In contrast to bilateral filtering, we are not interested in producing a weighted sample of local intensities but we instead consider the sum of the weights themselves which correspond to a robust estimate of the likelihood of $I_p$. One may further relate this to an information theoretic quantity of self-information in considering $-log(p(I_p))$, the self-information associated with the observation of intensity $I_p$.

With the above terms defined, Maxima in Information Scale-Space are defined as:

$$MISS(I_p) = \max_{\sigma_b} \left( -log(\sum_{q \in S} G_{\sigma_b}(||p - q||) G_{\sigma_i}(||I_p - I_q||)) \right) \qquad (2)$$

Saliency is therefore equated to the local self-information for the scale at which this quantity has its maximum value (for each pixel location) in a manner akin to scale selection based on normalized gradients or differential invariants [12]. This also corresponds to scale (and value) selection based on maxima in the sum of weights that define a local anisotropic diffusion process. In what follows, we comment further on conceptual connections to related work:

**1. Scale space extrema**: The definition expressed in equation 2 has a strong relationship to the idea of selecting extrema corresponding to normalized gradients in scale-space [1] or in curvature-scale space [13]. In this case, rather than a Gaussian blurred intensity profile scale extrema are evaluated with respect to local information expressed across scale space.
**2. Kadir and Brady**: In Kadir and Brady's proposal, interest points or saliency in general is related to the scale at which entropy is maximal [3]. While entropy and self-information are related, maxima in local entropy alone are insufficient to define salient content. Regions are therefore selected on the basis of the product of maximal local entropy and magnitude change of the probability density function. In contrast, the approach employed by MISS relies only on the expression in equation 2, and does not require additional normalization. It is worth noting that success in matching keypoints relies on the distinctness of keypoint descriptors which is a notion closely related to saliency.
**3. Attention based on Information Maximization (AIM)**: The quantity expressed in equation 2 is identical to the definition of saliency assumed by the AIM model [6] for a specific choice of local features, and a fixed scale. The method proposed in equation 2 considers the maximum self-information expressed across scale space for each local observation to determine relative saliency.
**4. Bilateral filtering**: Bilateral filtering produces a weighted sample of local intensity values based on proximity in space and feature space. The sum of weights in the normalization term provides a direct estimate of the likelihood of the intensity (or statistics) at the Kernel center, and is directly related to self-information.
**5. Graph Based Saliency and Random Walks**: Proposals for visual saliency also include techniques defined by graphs and random walks [14]. There is also common ground between this family of approaches and those grounded in information theory. Specifically, a random walk or Markov process defined on a lattice may be seen as a process related to anisotropic diffusion where the transition probabilities between nodes define diffusion on the lattice. For a model such as Graph Based Visual Saliency (GBVS) [14], a directed edge from node $(i, j)$ to node $(p, q)$ is given a weight

$w((i, j), (p, q)) = d((i, j)||(p, q))F(i - p, j - q)$ where $d$ is a measure of dissimilarity and $F$ a 2-D Gaussian profile. In the event that the dissimilarity measure is also defined by a Gaussian function of intensity values at $(i, j)$ and $(p, q)$, the edge weight defining a transition probability is equivalent to $W_p$ and the expression in equation 1.

## 3    Evaluation

In this section we present an array of results that demonstrate the utility and generality of the proposed saliency measure. This includes typical saliency benchmark results for both fixation prediction and object segmentation based on MISS. We also consider the relative invariance of this measure to image deformations (e.g. viewpoint, lighting) and demonstrate robustness to such deformations. This is accompanied by demonstration of the value of MISS in a more general sense in assessing saliency for a broad range of data types, with a demonstration based on 3D point cloud data. Finally, we also contrast behavior against very recently proposed models of visual saliency that leverage deep learning, revealing distinct and important facets of the overall problem.

The results that are included follow the framework established in section 2. However, the intensity value appearing in equations in section 2 is replaced by a 3D vector of RGB values corresponding to each pixel. $||.||$ denotes the L2 norm, and is therefore a Euclidean distance in the RGB colorspace. It is worth noting that the definition of MISS may be applied to arbitrary features including normalized gradients, differential invariants or alternative features. The motivation for choosing pixel color values is to demonstrate that a high level of performance may be achieved on standard benchmarks using a relatively simple set of features in combination with MISS.

A variety of post-processing steps are commonplace in evaluating saliency models, including topological spatial bias of output, or local Gaussian blur of the saliency map. In some of our results (as noted) bilateral blurring has been applied to the output saliency map in place of standard Gaussian blurring. The reasons for this are detailed later on in in this section, but it is worth stating that this has shown to be advantageous in comparison to the standard of Gaussian blur in our benchmark results.

Benchmark results are provided for both fixation data and salient object segmentation. For segmentation based evaluation, we apply the methods described by Li et al. [15]. This involves segmentation using MCG [16], with resulting segments weighted based on the saliency map [2].

### 3.1    MISS versus Scale

In considering scale space extrema, plotting entropy or energy among normalized derivatives across scale is revealing with respect to characteristic scale and regions of interest [3]. Following this line of analysis, in Figure 1 we demonstrate variation in information scale-space values as a function of $\sigma_b$ expressed in pixels. In Figure 1(a) three pixels are labeled corresponding to each of these categories as indicated by colored dots. The plot in Figure 1(b) shows the self-information for all of the selected pixels considering a wide range of scales. Object pixels, edge pixels and non-object pixels tend to produce different characteristic curves across scale in considering $-log(p(I_p))$.

### 3.2    Center bias via local connectivity

Center bias has been much discussed in the saliency literature, and as such, we include results in this section that apply a different strategy for considering center bias. In particular, in the following center bias appears more directly as a factor that influences the relative weights assigned to a likelihood estimate defined by local pixels. This effectively means that pixels closer to the center have more influence in determining estimated likelihoods. One can imagine such an operation having a more prominent role in a foveated vision system wherein centrally located photoreceptors have a much greater density than those in the periphery. The first variant of center bias proposed is as follows:

$$MISS_{CB-1}(I_p) = \max_{\sigma_b} \left( -log\left[ \sum_{q \in S} G_{\sigma_b}(||p - q||)G_{\sigma_i}(||I_p - I_q||)G_{\sigma_{cb}}(||q - c||) \right] \right) \text{ where, } c$$

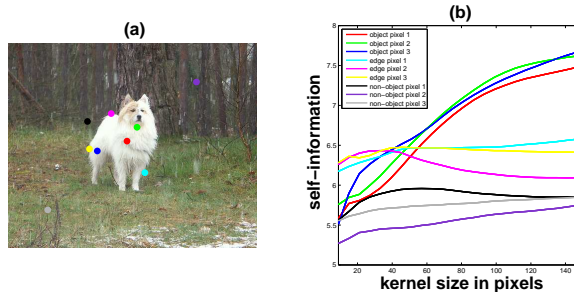

Figure 1: (a) Sample image with select pixel locations highlighted in color. (b) Self-information of the corresponding pixel locations as a function of scale.

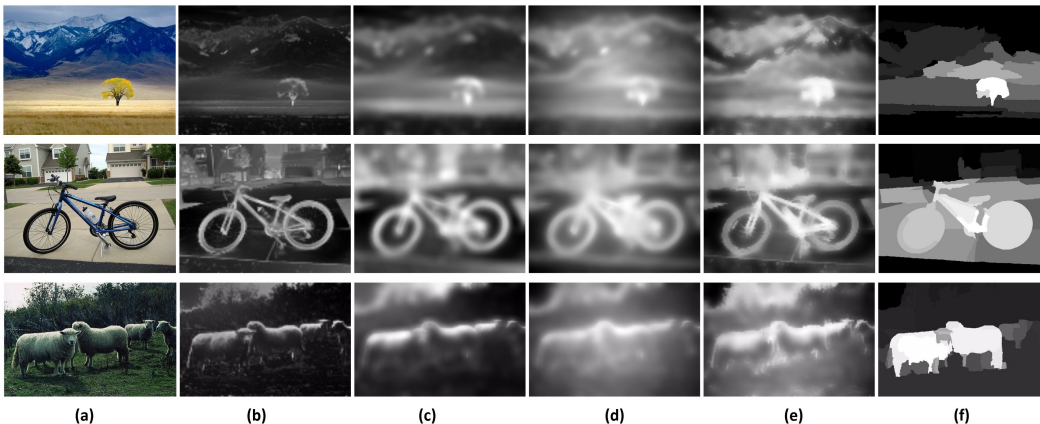

Figure 2: Input images in (a) and sample output for (b) raw saliency maps (c) with bilateral blur (d) using CB-1 bias (e) using CB-2 bias (f) object segmentation using MCG+MISS

is the spatial center of the image, $G_{\sigma_{cb}}$ is a Gaussian function which controls the amount of center bias based on $\sigma_{cb}$. The second approach includes the center bias control parameters directly within in the second Gaussian function.

$$MISS_{CB-2}(I_p) = \max_{\sigma_b} \left( -log\left[ \sum_{q \in S} G_{\sigma_b}(||p-q||)G_{\sigma_i}(||I_p - I_q|| \times (M - ||q-c||)) \right] \right) \text{ where,}$$

$M$ is the maximum possible distance from the center pixel $c$ to any other pixel.

### 3.3  Salient objects and fixations

Evaluation results address two distinct and standard problems in saliency prediction. These are fixation prediction, and salient object prediction respectively. The evaluation largely follows the methodology employed by Li et al. [15]. Benchmarking metrics considered are common standards in saliency model evaluation, and details are found in the supplementary material.

We have compared our results with several saliency and segmentation algorithms ITTI [18], AIM [6], GBVS [14], DVA [19], SUN [20], SIG [21], AWS [22], FT [23], GC [24], SF [25], PCAS [26], and across different datasets. Note that for segmentation based tests comparison among saliency algorithms considers only MCG+GBVS. The reason for this is that this was the highest performing of all of the saliency algorithms considered by Li et al. [15].

In our results, we exercise a range of parameters to gauge their relative importance. The size of Gaussian kernel $G_{\sigma_b}$ determines the spatial scale. 25 different Kernel sizes are considered in a range from 3x3 to 125x125 pixels with the standard deviation $\sigma_b$ equal to one third of the kernel width. For fixation prediction, only a subset of smaller scales is sufficient to achieve good performance, but the complete set of scales is necessary for segmentation. The Gaussian kernel that defines color distance $G_{\sigma_i}$ is determined by the standard deviation $\sigma_i$. We tested values for $\sigma_i$ ranging from 0.1 to 10. For

post processing standard bilateral filtering (BB), a kernel size of $9 \times 9$ is used, and center bias results are based on a fixed $\sigma_{cb} = 5$ for the kernel $G_{\sigma_{cb}}$ for CB-1. For the second alternative method (CB-2) one Gaussian kernel $G_{\sigma_i}$ is used with $\sigma_i = 10$. All of these settings have also considered different scaling factors applied to the overall image $0.25$, $0.5$ and $1$ and in most cases, results corresponding to the resize factor of $0.25$ are best. Scaling down the image implies a shift in the scales spanned in scale space towards lower spatial frequencies.

Table 1: Benchmarking results for fixation prediction

| s-AUC | aws | aim | sig | dva | gbvs | sun | itti | miss Basic | miss BB | miss CB-1 | miss CB-2 |
|---|---|---|---|---|---|---|---|---|---|---|---|
| bruce | 0.717[1] | 0.697[3] | 0.71[4] | 0.684 | 0.67 | 0.665 | 0.656 | 0.68 | 0.691[4] | 0.625 | 0.672 |
| cerf | 0.734[3] | 0.756 | 0.743[2] | 0.716 | 0.706 | 0.691 | 0.681 | 0.743[1] | 0.72 | 0.621 | 0.726[4] |
| judd | 0.829[2] | 0.82[4] | 0.812 | 0.807 | 0.777 | 0.806 | 0.794 | 0.807 | 0.809 | 0.832[1] | 0.825[3] |
| imgsal | 0.869[1] | 0.854 | 0.862 | 0.856 | 0.83 | 0.868[2] | 0.851 | 0.865[3] | 0.864[4] | 0.832 | 0.845 |
| pascal | 0.811[1] | 0.803 | 0.807[2] | 0.795 | 0.758 | 0.804[4] | 0.773 | 0.802 | 0.803 | 0.804[3] | 0.801 |

Table 2: Benchmarking results for salient object prediction (saliency algorithms)

| F-score | aws | aim | sig | dva | gbvs | sun | itti | miss Basic | miss BB | miss CB-1 | miss CB-2 |
|---|---|---|---|---|---|---|---|---|---|---|---|
| ft | 0.693[2] | 0.656[4] | 0.652 | 0.633 | 0.649 | 0.638 | 0.623 | 0.640 | 0.685[3] | 0.653 | 0.713[1] |
| imgsal | 0.595[1] | 0.536 | 0.590[2] | 0.491 | 0.557[4] | 0.438 | 0.520 | 0.432 | 0.521 | 0.527 | 0.582[3] |
| pascal | 0.569 | 0.587[1] | 0.566 | 0.529 | 0.529 | 0.514 | 0.585[2] | 0.486 | 0.508 | 0.583[3] | 0.574[4] |

Table 3: Benchmarking results for salient object prediction (segmentation algorithms)

| F-score | sf | gc | pcas | ft | mcg+gbvs [15] | mcg+miss Basic | mcg+miss BB | mcg+miss CB-1 | mcg+miss CB-2 |
|---|---|---|---|---|---|---|---|---|---|
| ft | 0.853[3] | 0.804 | 0.833 | 0.709 | 0.853[2] | 0.849[3] | 0.845[4] | 0.855[1] | 0.839 |
| imgsal | 0.494 | 0.571[2] | 0.612[1] | 0.418 | 0.542[3] | 0.535[4] | 0.513 | 0.514 | 0.521 |
| pascal | 0.534 | 0.582 | 0.600 | 0.415 | 0.675[2] | 0.667[4] | 0.666 | 0.679[1] | 0.673[3] |

In Figure 2, we show some qualitative results of output corresponding to MISS with different post-processing variants of center bias weighting for both saliency prediction and object segmentation.

### 3.4  Lighting and viewpoint invariance

Given the relationship between MISS and models that address the problem of invariant keypoint selection, it is interesting to consider the relative invariance in saliency output subject to changing viewpoint, lighting or other imaging conditions. This is especially true given that saliency models have been shown to typically exhibit a high degree of sensitivity to imaging conditions [27]. This implies that this analysis is relevant not only to interest point selection, but also to measuring the relative robustness to small changes in viewpoint, lighting or optics in predicting fixations or salient targets.

To examine affine invariance, we have to used image samples from a classic benchmark [5] which represent changes in zoom+rotation, blur, lighting and viewpoint. In all of these sequence, the first image is the reference image and the imaging conditions change gradually throughout the sequence. We have applied the MISS algorithm (without considering any center bias) to all of the full-size images in those sequences. From the raw saliency output, we have selected keypoints based on non-maxima suppression with radius = 5 pixels, and threshold = 0.1. For every detected keypoint we assign a circular region centered at the keypoint. The radius of this circular region is based on the width of the Gaussian kernel $G_{\sigma_b}$ defining the characteristic scale at which self-information achieves a maximum response. Keypoint regions are compared across images subject to their repeatability [5]. Repeatability measures the similarity among detected regions across different frames and is a standard way of gauging the capability to detect common regions across different types of image deformations. We compare our results with several other region detectors including Harris, Hessian, MSER, IBR and EBR [5].

Figure 3 demonstrates that output corresponding to the proposed saliency measure, revealing a considerable degree of invariance to affine transformations and changing image characteristics suggesting robustness for applications for gaze prediction and object selection.

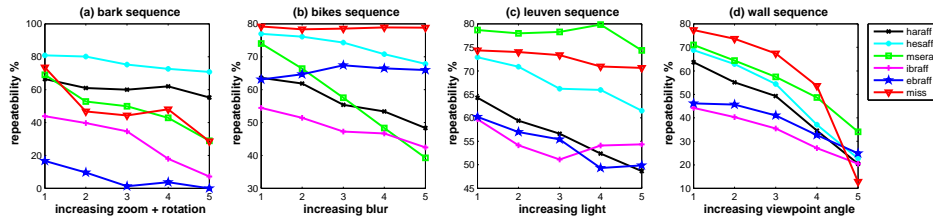

Figure 3: A demonstration of invariance to varying image conditions including viewpoint, lighting and blur based on a standard benchmark [5].

## 3.5 Beyond Images

While the discussion in this paper has focused almost exclusively on image input, it is worth noting that the proposed definition of saliency is sufficiently general that this may be applied to alternative forms of data including images/videos, 3D models, audio signals or any form of data with locality in space or time.

To demonstrate this, we present saliency output based on scale-space information for a 3D mesh model. Given that vertices are sparsely represented in a 3D coordinate space in contrast to the continuous discretized grid representation present for images, some differences are necessary in how likelihood estimates are derived. In this case, the spatial support is defined according to the $k$ nearest (spatial) neighbors of

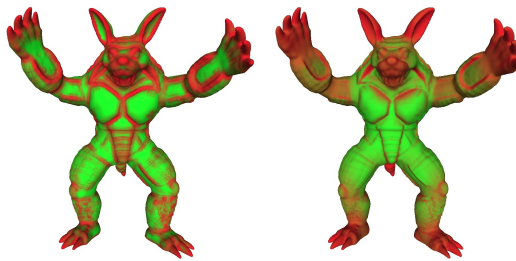

Figure 4: Saliency for 2 different scales on a mesh model. Results correspond to a surround based on 100 nearest neighbors (left) and 4000 nearest neighbors (right) respectively.

each vertex. Instead of color values, each vertex belonging to the mesh is characterized by a three dimensional vector defining a surface normal in the x, y and z directions. Computation is otherwise identical to the process outlined in equation 2. An example of output associated with two different choices of $k$ is shown in figure 4 corresponding to $k = 100$ and $k = 4000$ respectively for a 3D model with 172974 vertices. For demonstrative purposes, the output for two individual spatial scales is shown rather than the maximum across scales. Red indicates high levels of saliency, and green low. Values on the mesh are histogram equalized to equate any contrast differences. It is interesting to note that this saliency metric (and output) is very similar to proposals in computer graphics for determining local mesh saliency serving mesh simplification [28]. Note that this method allows determination of a characteristic scale for vertices on the mesh in addition to defining saliency. This may also useful to inferring the relationship between different parts (e.g. hands vs. fingers).

There is considerable generality in that the measure of saliency assumed is agnostic to the features considered, with a few caveats. Given that our results are based on local color values, this implies a relatively low dimensional feature space on which likelihoods are estimated. However, one can imagine an analogous scenario wherein each image location is characterized by a feature vector (e.g. outputs of a bank of log-Gabor filters) resulting in much higher dimensionality in the statistics. As dimensionality increases in feature space, the finite number of samples within a local spatial or temporal window implies an exponential decline in the sample density for likelihood estimation. This consideration can be solved in applying an approximation based on marginal statistics (as in [29, 20, 30]). Such an approximation relies on assumptions such as *independence* which may be achieved for arbitrary data sets in first encoding raw feature values via stacked (sparse) autoencoders or related feature learning strategies. One might also note that saliency values may be assigned to units across different layers of a hierarchical representation based on such a feature representation.

## 3.6 Saliency, context and human vision

Solutions to quantifying visual saliency based on deep learning have begun to appear in the literature. This has been made possible in part by efforts to scale up data collection via crowdsourcing in

defining tasks that serve as an approximation of traditional gaze tracking studies [31]. Recent (yet to be published) methods of this variety show a considerable improvement on some standard benchmarks over traditional models. It is therefore interesting to consider what differences exist between such approaches, and more traditional approaches premised on measures of local feature contrast. To this end, we present some examples in Figure 5 where output differs significantly between a model based on deep learning (SALICON [31]) and one based on feature contrast (MISS).

The importance of this example is in highlighting different aspects of saliency computation that contribute to the *bigger picture*. It is evident that models capable of detecting specific objects and modeling context are may perform well on saliency benchmarks. However, it is also evident that there is some deficit in their capacity to represent saliency defined by strong feature contrast or according to factors of importance in human visual search behavior. In the same vane, in human vision, hierarchical feature extraction from edges to complex objects, and local measures for gain control, normalization and feature contrast play a significant role, all acting in concert. It is therefore natural to entertain the idea that a comprehensive solution to the problem involves considering both high-level features of the nature implemented in deep learning models coupled with contrastive saliency akin to MISS. In practice, the role of salience in a distributed representation in modulating object and context specific signals presents one promising avenue for addressing this problem.

It has been argued that normalization is a canonical operation in sensory neural information processing. Under the assumption of Generalized Gaussian statistics, it can be shown that divisive normalization implements an operation equivalent to a log likelihood of a neural response in reference to cells in the surround [30]. The nature of computation assumed by MISS therefore finds a strong correlate in basic operations that implement feature contrast in human vision, and that pairs naturally with the structure of computation associated with representing objects and context.

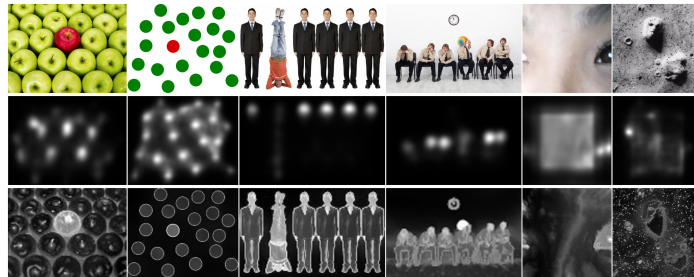

Figure 5: Examples where a deep learning model produces counterintuitive results relative to models based on feature contrast. Top: Original Image. Middle: SALICON output. Bottom: MISS output.

## 4  Discussion

In this paper we present a generalized information theoretic characterization of saliency based on maxima in *information scale-space*. This definition is shown to be related to a variety of classic research contributions in scale-space theory, interest point detection, bilateral filtering, and existing models of visual saliency. Based on a relatively simplistic definition, the proposal is shown to be competitive against contemporary saliency models for both fixation based and object based saliency prediction. This also includes a demonstration of the relative robustness to image transformations and generalization of the proposal to a broad range of data types. Finally, we motivate an important distinction between contextual and contrast related factors in driving saliency, and draw connections to associated mechanisms for saliency computation in human vision.

**Acknowledgments**

The authors acknowledge financial support from the NSERC Canada Discovery Grants program, University of Manitoba GETS funding, and ONR grant #N00178-14-Q-4583.

## Footnotes

[1] Although this example is based on pixels intensities, the same analysis may be applied to statistics of arbitrary dimensionality. For higher dimensional feature vectors, appropriate sampling is especially important.

[2]Note that while the authors originally employed CMPC [17] as a segmentation algorithm, more recent recommendations from the authors prescribe the use of MCG [16].

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
