[Supplementary Material]

# Supplementary Material for:
# Saliency, Scale and Information:
# Towards a Unifying Theory

**Shafin Rahman**
Department of Computer Science
University of Manitoba
shafin109@gmail.com

**Neil D.B. Bruce**
Department of Computer Science
University of Manitoba
bruce@cs.umanitoba.ca

## Abstract

This document contain supplementary information for the paper "Saliency, Scale and Information: Towards a Unifying Theory". This includes additional experiments, and details associated with the proposed algorithm.

## 1 Benchmarking Metrics

In this paper, the evaluation metrics follows the methods employed by Li et al. [1]. For completeness, a few additional definitions associated with benchmarking metrics, and associated parameter values are discussed.

**s-AUC:** Fixation data is characterized by spatial bias of fixation positions known as center bias. Center-bias creates a problem for design and evaluation of saliency algorithms in gauging performance differences due to algorithm behavior versus data bias. To solve this problem, Tatler *et al.* [2] proposed a shuffle AUC (sAUC) metric for evaluation. In this approach, standard ROC analysis is conducted with a change in how that data is sampled. Positive samples correspond to fixated locations for all observers for a given image and negative samples are randomly sampled from fixated locations across all other images in the dataset. By choosing the positive and negative samples based on the spatial distribution of the fixation data itself, this evaluation method effectively normalizes the effect of center bias.

**F-Score:** To evaluation segmentation performance we have used the F-score based on standard precision-recall analysis.

F-score $= (1 + \beta^2) \frac{precision.recall}{(\beta^2.precision) + recall}$ where, $\beta = \sqrt{0.3}$ used in our experiment (as in [3]).

**Segmentation model:** We have applied MCG [4] to produce object proposals for assigning saliency. To describe a proposal, we have used shape features (major axis length, eccentricity, minor axis length, the Euler number of the binary mask of the segment) and fixation distribution features i.e. a histogram of fixation densities for a corresponding object mask. The minimum and maximum area permitted for an object proposal are 200 and 400 pixels respectively. We have used 40% of the images for training and the rest of the images in testing. The random forest model is trained for classification using 16 trees. In the testing phase, each proposal is classified and the top 20 segments are used to generate final segmentation result. For a detailed description of this strategy, the reader is urged to consult with the work of Li et al. [1].

Additional qualitative results are presented in figure 2, which presents outputs for 3 images from each of the object segmentation datasets.

Figure 1: Impact of performance by changing $\sigma_i$ on (a) saliency and (b) segmentation dataset

## 2  Importance of $G_{\sigma_i}$ to output

The value of the standard deviation parameter $\sigma_i$ that defines luminance differences according to $G_{\sigma_i}$ has a significant impact on performance. This is the kernel bandwidth associated with a derived probability density estimate on intensities or local statistics. We have observed that this parameter is important for both saliency detection and salient object segmentation. This is visualized using wide range of $\sigma_i$ values as shown in Figure 1.

## 3  Non-Maxima Suppression

Non-maxima suppression for interest points is achieved using Kovesi's implementation [5]. This results in a selection of coordinates of peaks in the saliency map that exceed a given threshold, and suppression of super-threshold peaks of lower magnitude within a defined radius.

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

Figure 2: Input images in (a) and sample output for (b) raw saliency maps (c) with bilateral blur (d) using CB1 bias (e) using CB2 bias (f) object segmentation using MCG+MISS