[Reviews · NeurIPS 2015]

Submitted by Assigned_Reviewer_1

The paper proposes a simple, yet apparently powerful, saliency metric. For a given scale, it can be interpreted as the likelihood of the pixel intensity (or feature) under a distribution computed using kernel density estimation in a weighted neighborhood around the pixel. When using a Gaussian window for the neighborhood, the paper case the expression for the metric as being a scale-specific instantiation of a filtering operation. The final saliency of the pixel is then computed as the maxima of this per-scale metric across a range of scales. The paper provides an evaluation of this metric on a standard benchmark for predicting object saliency, both by itself as well as in combination with the MCG method, where it is found to perform competitively despite its simplicity. To showcase the possible application of the metric beyond images, a formulation for saliency on 3-D meshes is also introduced and discussed.

Overall, the paper is very well-written and does a good job of drawing parallels to a diverse set of methods in the literature that deal with saliency, segmentation, and the human visual system. The experiment results are compelling as well, and demonstrate that the metric has the potential to be practically useful. Overall, I am inclined to recommend acceptance.

I wonder if the authors could comment on possible extensions on more general formulations that expand the formulation beyond circularly symmetric windows at each scale. I wonder if there would be any benefit to generalizing the proposed metric to consider arbitrary 2-D Gaussian spatial windows, with the maxima now computed over general 2D covariance matrices instead of scalar scale values. Such a discussion might naturally fit into the section on viewpoint invariance ....
Summary: The paper proposes a new saliency metric for images, based on the maximal uniqueness of pixel intensity (or any per-pixel feature vector) as measured in neighborhoods at different scales.

Submitted by Assigned_Reviewer_2

How the MISS value is computed algorithmically and a study on the running time comparison between different saliency computation is appreciated.
Summary: The work is to propose a new saliency measure for images and others. While the central idea is not completely new, but is interesting as it unified several previous works on saliency computation. The work is acceptable for most conferences on image analysis or computer vision research, but could be in the borderline of NIPS publications.

Submitted by Assigned_Reviewer_3

This paper presents a new definition for visual saliency as maxima in information-scale space (MISS). Evaluation and comparisons are done on the tasks of fixation prediction and salient objection prediction. Robustness of MISS is characterized wrt

varying image conditions including: viewpoint, lighting and blur. MISS is also explained in terms of deep-learning-based saliency models.

Strengths: 1. Relates definition of saliency to scale-space information theory. 2. This new definition is naturally related to the problem of scale selection of local regions in both 2D images and meshes of 3D shapes. In computing saliency, scale can also be selected based on maximizing the information theoretic quantity of self-information.

Weaknesses:

1. Evaluation on the tasks of fixation prediction and salient object detection does not show overall better performance relative to the state of the art.

For example, see Table 1. MISS related approaches do not perform best on all datasets. In Table 2, only on the ft dataset, MISS performs the best. In Table 3, MISS performs the best only on the ft and PASCAL datasets.

2. Robustness to image transformations seem poor in Figure 3.

Summary: The paper presents a new definition for visual saliency as maxima in information-scale space (MISS), and relates it to a variety of previous works on scale-space theory, interest point detection, bilateral filtering and existing visual saliency models. However, somewhat weak evaluation results compared to other existing approaches put this work below the bar.

Submitted by Assigned_Reviewer_4

Summary:

The paper proposes a definition of visual saliency which is based on

Maxima in Information Scale-Space (MISS). Connections to scale-space theory and information theory are explained, as well as to some other works. Several experiments are reported, including on fixation and salient objects in 2D, and 3D salient features of 3D meshes.

Quality: The paper is well written, and it seems technically correct.

Connections between information theory and saliency are interesting (and to some extent have been explored in the past by others, although in somewhat different formulations), and combining these with scale-space theory makes a lot of sense (and, again, to some extent, has also been explored before; e.g., see below).

One problem with the quality of the paper is that the technical/mathematical contribution is fairly small, and pretty much boils down to a definition of a saliency measure.

Clarity:

The paper is well written.

Originality:

It is great the authors point out the connections to the classic works on scale-space theory, and tie together scale-space theory and information theory in the context of visual saliency. There is a paper, however, by Toews and Wells ("A mutual-information scale-space for image feature detection and feature-based classification of volumetric brain images", CVPRW, 2010) which seems to diminish *some* of the novelty of the present work. May the authors please clarify the relation to (and differences from) from Toews and Wells?

Significance: This paper may have some positive impact on a subset of the NIPS community.

Remark: it is always a pleasure (for me) to see a deep net fails (Figure 5 -- where the results obtained by the proposed method are clearly better than those obtained by the deep-net approach). However, perhaps the authors should add at least one counterexample? Surely there was at least one image where the SALICON output was better than the proposed method's output...

Summary: The paper proposes a new measure of visual saliency that is based on an interesting connection between the classical scale-space theory and information theory. Connections to few other works are nicely explained and the experimental section is sufficiently thorough. A main caveat about the paper is that the technical/mathematical contribution is rather small.

Submitted by Assigned_Reviewer_5

This paper proposes an information theoretic definition of visual saliency. The proposed approach is conveyed by building upon the scale space theory and nicely explained by drawing many connections to the existing literature.

An information theoretic view to the scale space studies already exist (for instance see the paper below), however applying it to the visual saliency problem is an interesting contribution.

Kristian Lindgren, "An Information-Theoretic Perspective on Coarse-Graining, Including the Transition from Micro to Macro", Entropy 2015

As a theoretical paper it is valuable. It gives a nice information theoretic insight for the saliency problem.

However in terms of the results it is below the state of the art. Nevertheless, the theoretical insights and the contribution deserves to be published.
Summary: The paper has some valuable theoretical contributions which are nicely explained in the context of existing literature. The experimental validation is not too strong, however the theoretical contribution puts the paper slightly above the acceptance threshold.

Author Feedback
Author rebuttal: We wish to thank the reviewers for their valuable feedback, and enthusiasm for this paper. There appears to be a broad appreciation for the theoretical contributions of this work, with a few questions concerning relationship to related work and interpretation of quality of benchmark results. We have addressed these concerns directly in the individual responses that follow:

R1: It is true that one might consider arbitrary 2-D Gaussian spatial windows based on a covariance matrix, in the same vein as affine invariant interest point selection. Both the underlying feature space and shape of support are factors that might be exercised to quantify saliency in a manner that is most advantageous for a particular application. It is also the case that even for the circular support, directional information may be derived from considering the kernel weights corresponding to the adaptive portion of a local bilateral kernel (that corresponds to the pixel-wise contribution to the local likelihood estimate). This directionality might be useful for efficiently converging on the optimal covariance defined window, but also more generally provides adjunct information for scene understanding, border ownership etc. than a scalar quantity defining saliency. We appreciate this suggestion, and plan to include discussion of these possibilities in the revised draft.

R2: The definition presented in Toews and Wells is very similar to that employed by Kadir and Brady. Entropy across scale is considered as in Kadir and Brady, but a penalty is incurred according to localizability (framed as mutual information in subtracting entropy across adjacent scales). This is in in the same spirit as the approach of weighting the entropy according to the derivative of the entropy-scale curve to select salient regions by Kadir and Brady. While entropy and scale are involved, this formulation does not carry the same generality as the proposal presented in the paper (or relationships to other research discussed). Nevertheless, this is an important reference that will be discussed in revisions. It is also worth mentioning that there are a number of advantages to the MISS proposal including the quality of output as a measure of saliency, the distinct cross-scale information profiles for different categories of image pixel selections, the generalization to mesh (and other) data, and directionality discussed in the response to R1. We will also include some examples that show a more favorable side of SALICON as suggested.

R3: The Lindgren paper relates information to scale, but is grounded in thermodynamics, state configurations and chemical systems. Any overlap therefore is mostly at a very coarse grained (or even superficial) level of abstraction, and does not diminish the value of any of the observations presented in our paper. Some of the value of our submission lies in relating a variety of distinct and important fields of inquiry, with many of the results competitive with or better than the current state-of-the-art in considering benchmarks. These are characteristics that are exclusive to our submission.

R5: There is some concern expressed that the approach does not perform best on metrics for *every* dataset. It remains the case that it is the best for many datasets, and we don't feel that that the value of the contribution rests on being best on every benchmark. The theoretical underpinnings of this work are of critical importance, and there are many cases where the approach is competitive with, or better than state-of-the-art despite of its simplicity. Moreover, the generality of what is proposed presents considerable opportunity for exploring model variants (e.g. alternate features) that do offer advantages for specific tasks. Regarding figure 3, this analysis is atypical for work involving saliency. The intent is not to outperform a long history of work in keypoint matching, but to reveal a striking degree of stability for a saliency model when using the proposed definition.

R6: We appreciate the encouraging remarks. We agree that this work presents an important contribution that is complementary to the current heavy emphasis on deep learning, and that also carries value in its simplicity and perspective on a large body of classic research.

R7: We intend to share the implementation for these methods, which will help to communicate the mapping from the mathematics to algorithmic form. The running times in general are implementation dependent, however given that the proposed method is highly parallelizable, we have written a very efficient GPU implementation. A more detailed analysis of running time (and complexity) are ongoing and will also be shared on the webpage dedicated to this work.